# Synergistic Antioxidant Effect of Prebiotic Ginseng Berries Extract and Probiotic Strains on Healthy and Tumoral Colorectal Cell Lines

**DOI:** 10.3390/ijms24010373

**Published:** 2022-12-26

**Authors:** Alessandra De Giani, Monica Oldani, Matilde Forcella, Marina Lasagni, Paola Fusi, Patrizia Di Gennaro

**Affiliations:** 1Department of Biotechnology and Biosciences, University of Milano-Bicocca, 20126 Milano, Italy; 2Department of Earth and Environmental Sciences, University of Milano-Bicocca, 20126 Milano, Italy

**Keywords:** prebiotics, ginseng, probiotics, functional food, nutraceuticals, oxidative stress, antioxidant effect

## Abstract

Oxidative stress caused by reactive oxygen species (ROS, O_2_•^−^, HO•, and H_2_O_2_) affects the aging process and the development of several diseases. A new frontier on its prevention includes functional foods with both specific probiotics and natural extracts as antioxidants. In this work, *Panax ginseng* C.A. Meyer berries extract was characterized for the presence of beneficial molecules (54.3% pectin-based polysaccharides and 12% ginsenosides), able to specifically support probiotics growth (OD_600nm_ > 5) with a prebiotic index of 0.49. The administration of the extract to a probiotic consortium induced the production of short-chain fatty acids (lactic, butyric, and propionic acids) and other secondary metabolites derived from the biotransformation of Ginseng components. Healthy and tumoral colorectal cell lines (CCD841 and HT-29) were then challenged with these metabolites at concentrations of 0.1, 0.5, and 1 mg/mL. The cell viability of HT-29 decreased in a dose-dependent manner after the exposition to the metabolites, while CCD841 vitality was not affected. Regarding ROS production, the metabolites protected CCD841 cells, while ROS levels were increased in HT-29 cells, potentially correlating with the less functionality of glutathione S-transferase, catalase, and total superoxide dismutase enzymes, and a significant increase in oxidized glutathione.

## 1. Introduction

Among the agents affecting human health, oxidative stress has aroused particular interest because it is known its involvement in aging and diseases progression [1,2]. At the basis of the oxidative stress there are the reactive oxygen species (ROS), including superoxide anion (O_2_•^−^), hydroxyl radical (HO•), and hydrogen peroxide (H_2_O_2_), which can damage cells outside and inside, notching proteins and nucleic acids [1]. Though, oxidative stress could be prevented and balanced through antioxidant defense systems, which comprehend bioactive compounds of different origins, both endogenous (human enzymes) and exogenous (bacteria and natural molecules) [3]. In the last category, functional foods mixing both specific probiotic bacterial strains and natural extracts are emerging as a real solution due to proven health benefits for immune system, intestinal microbiota, and brain through the gut-brain connection [2,4,5,6,7]. Indeed, among the probiotic features is counted the strain-dependent antioxidative ability. For instance, many scientific publications considered the versatile *Lactiplantibacillus plantarum* as an example of a highly scavenging bacterium, because of the inhibition of unsaturated fatty acids oxidation, the resisting hydrogen peroxide, the chelation of metal ions in vitro, and the capacity to restore liver function in vivo [3]. Other remarkable strains are *Lacticaseibacillus rhamnosus* GG [8], *Levilactobacillus brevis* BJ20 [9], and *Lacticaseibacillus casei* Zhang [10]. Therefore, the general health effects are related to the bacterium, its metabolites, and the responses induced in the host. Indeed, all the human cell types possess antioxidant enzymes belonging to the three major classes: catalases, superoxide dismutases, and glutathione peroxidases. Besides, there are a whole host of scavenger molecules such as glutathione and vitamins [11]. Remarkably, bioactive molecules with beneficial functions, including antioxidant ones, can also be introduced through the diet. Among the most famous, consumed, and ancient natural sources there is Ginseng, employed for over 2000 years in China as a traditional cure. Ginseng is part of the Araliaceae family, comprised in the *Panax* genus, and it is considered a perennial herb. Eleven species are known, but the most common and employed in the market are the Asian (*P. ginseng* Meyer), the North American (*P. quinquefolium* L.), and the Siberian (*Eleutherococcus senticosus*) [12,13]. Generally, the bioactive components are extracted from the roots and include polysaccharides, peptides, and ginsenosides [14]. Recently, other parts of the plant, previously considered as a waste, are also being analyzed, for example, the berries, and it is being discovered that they are also richer in bioactive molecules than the roots [15]. With respect to the active part, the ginsenosides are saponin molecules known to mediate most of the beneficial effects of the plant. Their extraction and analysis are challenging, anyway so far around 30 different ginsenosides have been extracted and chemically characterized [13]. Depending on the aglycone (i.e., “the non-sugar compound remaining after replacement of the glycosyl group from a glycoside by a hydrogen atom” [16]), they are categorized into protopanaxadiols (Rb1, Rb2, Rb3, Rc, Rd, Rh2, compound K, and Rg3) and protopanaxatriols (Re, Rf, Rg1, Rg2, and Rh1) [12,13]. The representatives of the protopanaxadiols Ra1-3, G-Rb1-2, and G-Rh2-3 are characterized by a dammarane backbone, while the protopanaxatriols Re, G-Rf, and G-Rg1 had an additional hydroxyl group on C6 in a dammarane backbone [17]. Noteworthy is the Re (C_48_H_82_O_18_) because it is a water-soluble compound mainly found in berries, leaves, flower buds, above all in *P. ginseng*, and it is known to improve immune response, skin protective barrier, antioxidant responses within cells, and reduce gastrointestinal motility dysfunction [13]. Recently, interest was also aroused by polysaccharides, able to exploit scavenging, antitumor, and anti-complementary activities [18,19]. Among the water-soluble sugars, the pectic polysaccharides, known as ginsan, include covalently bound domains of homogalacturonan (HG), and rhamnogalacturonan (RG)-I, II [15,20].

Thereby, given the characteristics of the probiotic strains, and the composition of Ginseng, scientists have begun to combine bacteria and Ginseng natural extracts to obtain enhanced effects on the host, particularly human hosts. Some studies focused the attention on the biotransforming capacities of *Bacillus subtilis*, *Aspergillus* subspecies, *Lactobacillus* subsp., and *Bifidobacterium longum* on ginsenosides resulting in deglycosylated molecules, for example, Re to Rg1 and Rg2, or Rb1 to Rd and compound K, still having their bioactive potential [21]. Others also investigated the prebiotic potential of Ginseng carbohydrates on specific probiotic strains (*L. plantarum* C88), inferring that acidic polysaccharides combined with *L. plantarum* prevented dangerous malondialdehyde generation, enhancing dose-dependently the scavenging activities of superoxide dismutase (SOD), glutathione peroxidase (GSH-Pox), catalase (CAT), and total antioxidant capacities (T-AOC) [18]. Finally, the gut microbiota modulation through intervention with Ginseng and probiotic strains was supposed to induce positive shifts in microbiome composition in healthy models (including in vitro reconstructed gut microbiota and in vivo models of mice and humans), resulting in improved intestinal immunity and metabolism [22]. For example, in an in vitro experiment employing a microbiota derived from human feces, it was noted that there was a modulation in the levels of Proteobacteria and Bacteroidetes, in particular of bacteria belonging to the genera *Escherichia*, *Streptococcus*, *Ruminococcus*, *Dorea*, and *Prevotella* [23]. While administering Korean ginseng to a healthy mouse model, Han et al. [24] noted an increment in the total bacterial count, specifically of Lactobacilli. Moreover, the release of probiotic’s beneficial secondary metabolites short-chain fatty acids (SCFAs) was implemented in the presence of ginsenosides, suggesting that saponin molecules could be directed in the metabolic pathways leading to SCFAs production [22]. Therefore, a synergism between the host and its own intestinal microbiota could be possible, thinking about an eubiotic microbiota that makes biomolecules (such as ginsenosides) even more available, and gives the colonocytes additional beneficial molecules (SCFAs) and protection from oxidative stress.

This scientific article aims to describe the production and characterization of an effective extract from *P. ginseng* in terms of beneficial activities on the host. Beneficial effects include the preferential stimulation of different probiotics growth and the subsequent production of SCFAs, and other secondary metabolites derived from the biotransformation of Ginseng components (evaluated as a prebiotic capacity). On the other hand, the direct effects for the host are evaluated by administering the Ginseng berries extract or the produced secondary metabolites after probiotic fermentation directly on healthy and colorectal cancer cell lines, evaluating the cell viability and the response to the induced oxidative stress, based on the evaluation of the general response to this kind of stress, the involvement of the most effective scavenging enzymes, and the glutathione action.

## 2. Results

### 2.1. Determination of Panax Ginseng Berries Extract Principal Molecules

The main representative classes of molecules of *P. ginseng* berry extract are listed in Table 1. The most abundant components are pectin-based polysaccharides (54.30%). Galactose molecules represent 9.01%, while the galacturonic acid-like molecules amount to 4.84%. Other identified molecules belong to the category of ginsenosides (10.00%) and ginsenosides Re (2.00%). Finally, the presence of proteins and polyphenols is low (0.98% and 0.72%, respectively). The remaining 34.99% is composed of unidentified molecules.

### 2.2. Evaluation of the Prebiotic Potential of Panax Ginseng Berries Extract

The possible capability of *P. ginseng* berries extract to improve the growth of several probiotic bacteria of the genera *Lactobacillus* and *Bifidobacterium* was tested at 2% *w*/*v*. As shown in Figure 1, all the challenged probiotics could ferment the Ginseng extract and the differences are statistically significant comparing the values to the growths of control cells (*p*-value < 0.0001 and < 0.001). Considering *Lactobacilli*, *L. fermentum* showed the most positive responses (Optical Density at 600 nm (OD_600nm_) > 5, *p*-value < 0.0001); while, regarding the probiotics belonging to Bifidobacteria, *B. longum* subsp. *infantis* reached the highest growth (OD_600nm_ > 6, *p*-value < 0.001). Nevertheless, the other strains also grew well on the extract (*p*-value < 0.0001, < 0.001, and < 0.01).

Concerning possible commensal members of the intestinal community (i.e., *B. cellulosilyticus*, *B. finegoldii*, *F. plautii*, *C. symbiosum*, *E. coli*, and *R. gnavus*), it is interesting that the 2% *w/v* Ginseng extract sustained their growth, but not reaching the same OD_600nm_ values of the probiotic strains (Appendix A).

To support the possible prebiotic potential of *P. ginseng* extract, the prebiotic index (PI) was calculated. The highest PI value is the one of the Ginseng extract, corresponding to 0.49, while the control is around zero (0.08). 

Altogether, the results support the possibility to consider the Ginseng berries extract as a prebiotic source in the intestinal milieu.

### 2.3. Investigation of the Capacity of the Single Components of Panax Ginseng Berries Extract to Support Probiotics Growth

Having regard to the positive data concerning the growth of the probiotics on the whole Ginseng berries extract, it was investigated which dosed component could most support bacterial growth. Therefore, the eight strains were grown in presence of pectin-based polysaccharides, galactose, and galacturonic acid. As shown in Figure 2, not all bacteria utilized the molecules in the same way. As expected, the simplest molecule galactose is the most consumed substrate for growth; indeed, *L. plantarum* (*p*-value < 0.001, vs. mMRS), *L. reuteri* (<0.001), *B. longum* subsp. *infantis* and B. *longum* (<0.05) are those who have reached the most significant OD_600nm_ values. Then, the second most fermented molecule is pectin, though only the *Lactobacillus* strains *L. plantarum* (*p*-value < 0.01, vs. mMRS), *L. reuteri* (<0.01), and *L. rhamnosus* (<0.05) are the most efficient in using it.

Curiously, *B. animalis* subsp. *lactis* could not grow on galactose and galacturonic acid (Figure 2).

### 2.4. Growth of a Beneficial Probiotic Mix on Panax Ginseng Berries Extract

To understand if a mix of probiotic bacteria could interact as a consortium and better support the beneficial effects on the host, previously tested probiotic strains (Table 2) were combined and tested on Ginseng berries extract. As reported also in De Giani et al. [5], they were homogenized in a correct ratio, and then inoculated at OD_600nm_ equal to 0.1. The capacity to utilize the natural extract to grow was assayed under an experimental set up identical to the one employed with the individual strains, i.e., the minimum mMRS supplemented with 2% *w*/*v* of Ginseng berries extract. OD_600nm_ were recorded at the end of the anaerobic fermentation (Figure 3). The probiotic consortium showed a growth value of OD_600nm_ 3.98 ± 0.06 and it is a significantly high growth compared to the one on the control medium (*p*-value < 0.001). Therefore, the potential prebiotic effect of Ginseng berries extract is also confirmed in this type of experiment.

### 2.5. Analyses of Probiotic Secondary Metabolites in Presence of Panax Ginseng Berries Extract

The investigation of the metabolites produced by the probiotic consortium exploited an ethyl acetate extraction of the culture broth, carried out after 48 h of anaerobic fermentation at 37 °C on *P. ginseng* berries extract, subsequently analyzed by GC-MSD.

With respect to the control condition, samples deriving from Ginseng berries extract fermentation by the probiotic consortium showed complex chromatograms, with numerous new peaks (Figure 4). The main one, at a retention time (R_t_) of 11.2 min, is associated with butyrate. The second most abundant is at R_t_ of 5.7 min, and it is identified as cyclopentane, which is an alkaloid probably present in the Ginseng berries extract. Then, the third highest peak is lactic acid (R_t_ of 7.8 min). In addition, acetic (retention time of 5.3 and 6.7 min), thiocyanic (R_t_ of 8.1 min), propionic (R_t_ of 8.7 min), valeric (R_t_ of 10.8 min), and oxovaleric (R_t_ of 12.6 min) acids were identified.

### 2.6. Evaluation of the Bioactivity of Extracted Secondary Metabolites on Cell Viability

Through an MTT viability assay, it was possible to observe whether the total Ginseng berries extract (ET), the fermented Ginseng berries extract from the probiotic consortium (EF), or mMRS medium fermented by the probiotic consortium (mMRS) could influence the cell viability of the healthy and tumor cell lines. Three concentrations for each extract (0.1, 0.5, and 1 mg/mL) were tested. Results reported in Figure 5 show that ET had slight dose-dependent toxicity for the CCD841 cell line, although the decrease in cell viability was not statistically significant. This trend, on the contrary, was absent in cells treated with EF, which appears to be less toxic than the previous one. The HT-29 cell line, on the other hand, shows a clear dose-dependent sensitivity to treatment only when the cells were incubated with EF and not with ET. The 1 mg/mL dose of mMRS is statistically harmful to both these cell lines. Based on these results, it was decided to proceed with the subsequent experiments by choosing to test only the 0.5 mg/mL concentration of both extracts (ET and EF). This concentration was found to be discriminating for testing the effects induced by ET and EF in both cell lines without incurring the toxicity of the nutrient solution (mMRS). At this concentration, EF was only toxic to the tumoral cells, while the viability of healthy cells remained similar to that of the untreated control.

### 2.7. Oxidative Stress Analysis

Both cell lines were treated with Ginseng berries extract and the derived secondary metabolites after probiotics fermentation; then, the levels of oxidative stress to which the cells could be subjected were analyzed. The experiments were based on the ability of specific probes to become fluorescent after coming into contact with their substrates. In particular, the H_2_DCF-DA probe was used to detect the presence of generic reactive oxygen species, while the fluorescence of DHE and MitoPY indicators was utilized for evaluating the level of superoxide anion or mitochondrial hydrogen peroxide, respectively. 

ET did not cause any significant change in ROS levels in both healthy (Figure 6A) and tumor cells (Figure 6B). On the other hand, treatment with EF considerably reduced the presence of generic ROS and of hydrogen peroxide in healthy cells (Figure 6A); while in the tumor cell line, an increase in the fluorescence of all three probes can be observed, suggesting an increase in ROS, superoxide anion, and hydrogen peroxide levels (Figure 6B). 

To understand the decrease of the oxidative stress in healthy cells or its increase in tumor cells, the action of GST, GR, GPox, CAT, and SOD tot (Figure 6C,D) as representatives of enzymes involved in the antioxidant functions, and the levels of glutathione were tested. Indeed, a change in the level of antioxidant enzyme activity can be considered a sensitive biomarker of cellular response to oxidative stress [25]. 

For the CCD841 cell line, an improvement of the enzyme activity of GST, GPox, and CAT was measured both when the cells were treated with ET and EF, while the GR activity increased only after treatment with EF. SOD tot did not modify its detoxifying action under any conditions (Figure 6C). In the HT-29 cell line, the most marked effects were found after treatment with EF. In this case, all enzymes, except a slight decrease in GPox, showed a significant change in their activity (Figure 6D). GST, CAT, and SOD tot enzymes are less functional, while GR increases its effectiveness compared to control cells. In the condition of HT-29 cells treated with ET, a reduction in GPox activity and an increment in CAT were observed. 

Finally, the levels of GSH tot, GSH, and GSSG present in the cells after the treatments were analyzed. In healthy cells, the treatment with ET does not bring any variation in the levels of each form of glutathione; meanwhile after EF treatment, a decrease in all the values analyzed compared to the control was highlighted. A slight increase in GSSG in CCD841 treated with EF led to the decrement of GSH/GSSG ratio (Figure 6E, Appendix A). For the tumor cell line, Figure 6F displays a substantial decrease in GSH after ET treatment, while a significant increment in the oxidized form and a slight increase in total glutathione occur when the cells have been treated with EF.

## 3. Discussion

From the literature and the Chinese medicine, the beneficial effects of the various extracts deriving from various structures of the Ginseng plant, including berries, are linked to the immune system, blood circulation, and brain activity [9,13,17,26] because of the high presence of ginsenosides [15]. Nowadays, the use of berries is valuable given the circular economy and the reduction of waste with possible bioactive molecules; indeed, berries are mainly considered a by-product of the ginseng root culture [22]. 

In this work, we used berries from *Panax ginseng* C.A. Meyer manually harvested and then subjected to initial shredding operations. The bioactive molecules and the polysaccharides were extracted from the natural matrix using 60% aqueous methanol, instead of 70% methanol solution [12]. After the preliminary characterization, the most abundant constituents of our extract resulted in pectin-based polysaccharides, followed by molecules belonging to ginsenosides and ginsenosides Re. The concentration of proteins and polyphenols is drastically lower. The results are in line with other quantifications of Ginseng berries extracts described in the literature. For example, in Cho et al. [27] the water-soluble sugars varied between 23.2 and 49.5%, while the acidic polysaccharides were 2.9 to 6.9%, and the total polyphenols were around 0.5%. Interestingly, we obtained more total ginsenosides with respect to Cho et al. [27], indeed we can account for about 10% of the bioactive molecules vs. 2%. We measured about 2% of Re, similarly to Gao et al. [13], describing 3.5% of ginsenosides Re in berries of American *P. ginseng*. Finally, the non-identified component, corresponding to about 34%, could be starch because it is reported that starch accounts for 20–30% of Ginseng [19]. Although in the scientific literature there is not much information on berries, the composition of our extract is in line with that of the ginseng root, too [28].

Consequently, we investigated the possible prebiotic potential of the Ginseng berries extract, evaluating its efficacy in boosting the growth of different probiotics of the *Lactobacillus* and *Bifidobacterium genera*. From obtained data, we could speculate that the extract could be considered a prebiotic because all the probiotics were able to ferment it and the differences between the control and the medium added with the natural extract were statistically significant. Among the Lactobacilli, *L. fermentum* was the strain that grew the most, while *B. longum* subsp. *infantis* reached the highest OD_600nm_ value as regards the Bifidobacteria. The growth is comparable to the same strains on Maitake extract at the concentration of 2% *w*/*v* [5], though they reached final OD_600nm_ values between 3 and 4, while on the Ginseng the values were higher (between 4 and 6). Furthermore, the growth capacity of these probiotics on a recognized prebiotic source, namely FOS from chicory with a 3 to 5 degree of polymerization at 2% *w*/*v* [29], are comparable. However, also the bacteria belonging to the basal community strains (i.e., *B. cellulosilyticus*, *F. plautii*, *C. symbiosum*, *E. coli*, *R. gnavus*, and *B. finegoldii*) showed a positive response after the exposition to the Ginseng berries extract. Therefore, to strongly assess the prebiotic potential of the extract, we calculated the PI. The obtained value of 0.49 suggested a slight prebiotic potential and it is in line with the values of highly methylated pectin [30]. To better understand which carbohydrate component could support bacterial growth, the single polysaccharides composing the extract, i.e., pectin, D-galacturonic acid, and galactose were tested. Pectin is composed of D-galacturonic acid, L-rhamnose, D-arabinose, D-galactose, and other monosaccharides [31]. The smooth regions of pectin are principally composed of homogalacturonanas (HG), which are formed by D-galacturonic acids linked by α-1,4 glycosidic bonds. HG esterification with methanol is dependent on the botanical origin, the part of the plant, the climate, and the extraction method employed. It is considered a new-generation prebiotic because of recent data regarding the promotion of helpful microorganisms within the gastrointestinal system [32]. Among the tested strains, the most positive results were attributed only to Lactobacilli (*L. plantarum*, *L. reuteri*, and *L. rhamnosus*). While on D-galactose also *B. longum* subsp. *infantis* and B. *longum* showed to be sustained by the molecule. The results are of interest because the fermentation ability is linked to the defence from the oxidative stress prompted by aging, in particular in combination with *L. plantarum*, *B. longum*, and *B. animalis* [2,3].

To enhance the beneficial outcomes due to the concurrent administration of prebiotics and probiotics, the growth of a beneficial probiotic mix in the presence of Ginseng berries extract was analyzed. The prebiotic effect of the Ginseng was confirmed, and the growth was statistically significant compared to the control condition. Therefore, we proceeded with the extraction and characterization of the secondary metabolites produced, as it is another necessary condition to define a molecule as a prebiotic. Gas chromatographic analyses highlighted that the samples deriving from the fermented Ginseng extract presented a complex chromatogram with respect to the control condition. Acids associated with the growth of lactic bacteria (LAB) have been identified, such as lactic, propionic, butyric, and acetic acid [5,18]. These metabolites have a remarkable range of beneficial properties for colon health. Besides being the preferential source of colonocyte energy, they maintain mucosal integrity, reduce pro-inflammatory cytokines and provoke apoptosis in colorectal cancer cell lines [33]. Instead, oxovaleric and thiocyanic acids are secondary metabolites produced mainly by bacteria belonging to the genus *Lactobacillus*, in particular from *L. rhamnosus* [34]. Finally, cyclopentane is characterized by the typical ring of steroidal alkaloids such as ginsenoside Rg1 [35]. 

Given the metabolites, we evaluated their effects on healthy and cancerous intestinal cell lines. Primarily, by the MTT assay, the vitality of healthy CCD841 and HT-29 tumoral cell lines was determined. The choice fell on this tumoral cells because they are an excellent model for the study of colorectal cancer, as well as the most used in studies related to food. After all, HT-29 express the characteristics of mature intestinal cells [36]. Both the cell lines were exposed to increasing concentrations of total Ginseng berries extract (ET), the fermented Ginseng berries extract from the probiotic consortium (EF), and mMRS fermented by the beneficial probiotic mix (mMRS). None of the tested samples influenced CCD841, while HT-29 showed a dose–response effect to EF. The subsequent experiments were set up to investigate the levels of ROS and the related cellular responses, due to the known antioxidant potential of Ginseng and the potentiated effect derived from its fermentation by probiotics [18,21]. This is in accordance with former results obtained with the employed probiotic strains, characterized for the antioxidant effect on BALB/c3T3 cells [4]. Both CCD841 and HT-29 were treated with ET, EF, or mMRS at a single concentration of 0.5 mg/mL because it was the only one effective for the discrimination between the two cell lines without being lethal. The levels of oxidative stress to which the cells could be subjected were analyzed by specific probes for generic ROS, O_2_•^−^, and mitochondrial H_2_O_2_. Interestingly, only EF induced a decrease of ROS, specifically H_2_O_2_ in the healthy cell line, whereas in the tumor one a general increment of all the evaluated ROS was registered. A similar protective effect was assessed by Jung et al. [21], which described the conversion of ginsenosides Rb2 to Rb3 by *L. plantarum* KCCM 11613P, enhancing the antioxidant effect of the Red Ginseng extract. Furthermore, He et al. [18] reported that *P. ginseng* polysaccharides along with *L. plantarum* C88 demonstrated significant DPPH, ABTS, and superoxide anion radicals scavenging effects, suggesting the implication of SOD, Gpox, and CAT. Accordingly, we decided to test the specific activities of these enzymes, and other main actors involved in ROS detoxification (GST, GR, Gpox, CAT, and SOD). The significant effects were highlighted in the HT-29 cell line after the treatment with EF. Indeed, all enzymes were less functional, while GR increases its effectiveness with respect to the control. Instead, in the healthy cell line the enzymatic response increased both in presence of ET and EF. These data reflect in vivo results of Ginseng polysaccharides combined with C88 strain, which increased the SOD, Gpox, and CAT activities in serum and liver in the mice-treated group, with respect to the aging control group [18]. To further investigate the GR increase in cells treated with EF, the intracellular level of total glutathione was measured, quantifying both its reduced and oxidized form, to also calculate their ratio. A moderate increment in GSSG in CCD841 has led to a reduction of GSH/GSSG, while a significant increase in the oxidized form and a slight increase in total glutathione occurs when the HT-29 were treated with EF. There is likely an increment in oxidative stress, linked to a need for greater availability of scavenger molecules. This could correlate with the increase in GR activity. In this scenario and consistent with our findings, it is proposed that natural molecules and LAB could have a protective effect on oxidative stress, thanks to the modulation of the main antioxidant enzymes [18,37,38].

In conclusion, results obtained in this work confirm that the *P. ginseng* C.A. Meyer berries extract shows prebiotic effects. The metabolites obtained from the fermentation of the extract do not appear harmful to the healthy cells but affect the tumoral cell line. Furthermore, the impact on the antioxidant network is evident considering the HT-29, with the increase in oxidative stress and, consequently, an increase in the scavenging activity aimed at counteracting it. Therefore, a synbiotic composed of beneficial probiotic strains and the Ginseng berries extract could be proposed as an adjuvant in an anticancer drug against colorectal cancer or in the protection against the free radicals naturally occurring during the aging process. Furthermore, the extraction performed from berries and not from the roots could be an innovative approach, to the recovery of materials previously considered waste.

## 4. Methods and Materials

### 4.1. Ginseng Berries Extract Production

The *Panax ginseng* berries extract (*P. ginseng* C.A Meyer, sold as “Panaxolyde”) was supplied by Flanat Research Italia Srl (Rho, Italy). The origin of the plants is Asian, and they are cultivated in wild-simulated farming. They were manually harvested in the period between September and October. The berries were chosen for the preparation of the extract, as follows. Firstly, berries underwent grinding and weighing. Then, the obtained material was subject to ethanol:water (Sigma, Milano, Italy) extraction, using a drug-extract ratio of 40:1. After separation phase, the harvested sedimented material was concentrated, and put at 50 °C to completely evaporate the ethyl alcohol. The resulting substance was subject to blending and sieving, having a fine powder qualified by brown color and characteristic odor.

### 4.2. Description of the Components of Ginseng Berries Preparation

#### 4.2.1. Determination of the Protein Content

The Ginseng extract protein concentration was evaluated by the Bradford method [39]. Bovine serum albumin (BSA, Sigma, Milano, Italy) was used to construct the calibration curve, in a range between 0 and 1 mg/mL. The sample (Ginseng powder) was instead tested at 1 mg/mL and 10 mg/mL. The reaction mixture was prepared directly in the cuvette by adding 1.5 mL of Bradford’s reagent (Sigma, Milano, Italy) and 50 μL of the sample (BSA for the calibration curve, Ginseng as sample, H_2_O as control). It is left to incubate for 1 min at ambient temperature, then the concentration was read as absorbance at 595 nm. All samples were analyzed in triplicate.

#### 4.2.2. Determination of Polyphenol Content

The Folin–Ciocalteau phenol test [40] was used to evaluate the polyphenol concentration. The calibration line was set up with gallic acid (GA, Sigma, Milano, Italy) as standard, increasing the concentrations from 0 to 100 μg/mL. The sample (Ginseng powder) was tested at two different concentrations as before. The reaction mixture was prepared directly in the cuvette following the indication of the manufacturer (Sigma, Milano, Italy). After 30 min of incubation at room temperature, the absorbance at 760 nm has been registered. All samples were analyzed in triplicate.

#### 4.2.3. Determination of Pectin-Based Polysaccharides, Galactose, and Galacturonic Acid Content

To quantify the content of pectin-based polysaccharides, the phenol-sulfuric acid assay [41] known for the determination of sugars and related substances was employed. Standard pectin (Pectin from apple, poly-D-galacturonic acid methyl ester, Sigma, Milano, Italy) with a concentration between 0 and 100 μg/mL was employed for the standard curve. The sample was tested at two different concentrations as described previously. The reaction mixtures were prepared in glass tubes and left to incubate for 15 min in a thermostatic bath at 30 °C. The absorbances at 490 nm were recorded. All specimens are analyzed three times.

The same protocol was utilized to obtain the content of free galactose. The calibration curve was constructed in the same way as previously described with a D-galactose solution (D-(+)-Galactose, Sigma, Milano, Italy).

Galacturonic acid-based molecule concentration was evaluated by a modified Anthon et al. [42] assay. A copper buffer was made by mixing 2.32 g of NaCl, 0.32 g of NaOAc, and 0.1 mL of glacial CH_3_COOH to 8 mL of water. Then, 0.05 g copper (II) sulphate was included, and the pH was brought to 4.8 using 1M sodium hydroxide, reaching 10 mL final volume using water. All the reagents employed were from Sigma, Milano, Italy. To construct the calibration curve, a standard solution of D-galacturonic acid (D-(+)-Galacturonic acid monohydrate, Sigma, Milano, Italy) between 0 and 500 μg/mL was used. The sample was tested at two concentrations of 1 and 10 mg/mL. The reaction mixture was prepared in glass tubes by adding 100 μL of the sample (D-galacturonic acid for the calibration curve, Ginseng as sample, H_2_O as control) and 100 μL of buffered copper solution. The tubes are covered and positioned in 100 °C thermostatic bath for 40 min. Then, they were cooled to room temperature and 200 μL are moved to a spectrophotometer cuvette. 800 μL of the 2N Folin-Ciocalteau reagent (Sigma, Milano, Italy) diluted 1:40 in water was thrown in. Immediately the samples became colored, and their absorbance was measured at 750 nm. All samples were analyzed three times. 

#### 4.2.4. Measurement of Ginsenosides Amount

The determination of the total ginsenosides and the ginsenosides Re content was performed using an HPLC method as described by Brown and Yu [12]. The HPLC used was a Thermo Finningan Surveyor Plus HPLC apparatus (Thermo Fischer Scientific, Waltham, MA, USA) equipped with a Gemini C18 analytical column (150 × 2.0 mm i.d., 5 μm, Phenomenex, Torrance, CA, USA).

### 4.3. Bacteria Employed and Their Maintenance

In Table 2 are listed the bacteria utilized in this work. Probiotic *Lactobacillus* and *Bifidobacterium* [4] were from the private collection of Roelmi HPC (Origgio, Italy). The basal community strains were provided by BEI Resources (Manassas, VA, USA), NIAID, NIH, as part of the Human Microbiome Project: *Bacteroides cellulosilyticus*, strain CL02T12C19, HM-726; *Clostridium orbiscindens*, strain 1_3_50AFAA, HM-303 (formerly *Flavonifractor plautii*); *Clostridium symbiosum* WAL-14673, HM-319; *Bacteroides finegoldii*, strain CL09T03C10, HM-727; *Ruminococcus gnavus*, strain CC55_001C, HM-1056. The strain *Escherichia coli* ATCC 25922 is from the American Type Culture Collection (ATCC, Manassas, VA, USA). 

Normally, Lactobacilli and Bifidobacteria were maintained in liquid De Man, Rogosa and Sharp medium (MRS) (Conda Lab, Madrid, Spain) complemented with 0.03% L-cysteine (Sigma, Milano, Italy) for 2 days. Instead, the basal community strains were grown on Reinforced Clostridia Medium (RCM) (Conda Lab, Madrid, Spain), supplemented with 0.03% L-cysteine, and 0.01 g/L of hemin (Sigma, Milano, Italy) [43] for 72 h. All the strains grow at 37 °C, in anaerobiosis generated by Anaerocult GasPack System (Merck, Darmstadt, Germany).

For the growth assays, the modified MRS (mMRS, [44]) plus 0.03% L-cysteine, was employed as a basal medium. The pH was brought to 6.8 with 6M sodium hydroxide (Sigma, Milano, Italy) before sterilization (121 °C for 20 min). Ginseng berries extract, pectin (Pectin from apple, poly-D-galacturonic acid methyl ester, Sigma, Milano, Italy), galactose (D-(+)-Galactose, Sigma, Milano, Italy), and galacturonic acid (D-(+)-Galacturonic acid monohydrate, Sigma, Milano, Italy) were supplemented individually to mMRS as sole carbon sources (2%, 1.08%, 0.18%, and 0.1% *w*/*v*, respectively). All the substrates were dissolved in MilliQ water and 0.22 µm filtered (Merk Millipore, Darmstadt, Germany). Regarding pectin, it was allowed the hydration of the fiber for 1 overnight before use.

### 4.4. Growth Assays of Single Bacteria or Beneficial Probiotic Mix

Before the setup of the experiments, Lactobacilli and Bifidobacteria (Table 2) were pre-cultivated anaerobically for 2 days, at 37 °C, in MRS plus 0.03% L-cysteine, while the basal community strains (Table 2) were pre-cultivated in RCM added with 0.03% L-cysteine, and 0.01 g/L of hemin for 72 h, at 37 °C in anaerobiosis.

The medium for the single strain growth assays was described previously (in Section 4.3), while the method is described in De Giani et al. [45].

Tests with beneficial probiotic mix [5] were conducted in a volume of 10 mL in a plastic tube (CLEARLine, Venezia, Italy). Final OD_600nm_ was measured after an incubation period of 48 h, in anaerobic conditions.

### 4.5. Determination of Produced Bacterial Secondary Metabolites

#### 4.5.1. Metabolites Extraction

The evaluation of the production of metabolites, i.e., SCFAs and by-products of Ginseng berries extract metabolism by the beneficial probiotic mix, was developed after the fermentation period as reported in De Giani et al. [5], with slight variations. Firstly, cultures were subjected to 7000 rpm centrifuge (Eppendorf, Milano, Italy) for ten minutes at 20 °C. Then, after separation of cell pellet from the broth, the supernatant was brought to pH 2 using 6 M chloride acid (Sigma, Milano, Italy). An equal volume of ethyl-acetate (Sigma, Milano, Italy) was used for the strong manual agitation extraction for 20 min. After the separation of the two phases, the organic phase was withdrawn, and kept. To the remaining liquid, other ethyl-acetate was added in a ratio of 1:1, followed by other 5 min of mixing. After phase separation, the organic one was grouped with the first one.

#### 4.5.2. Extracted Metabolites Analyses 

In De Giani et al. [5] is reported the protocol for gas-chromatographic analyses (Technologies 6890 N Network GC System), connected to 5973 Network Mass Selective Detector (Agilent Technologies, Santa Clara, CA, USA). Little variations in instrument configuration were applied. The capillary column used was a J&W DB-5ms Ultra Inert GC Column (fused silica, 60 m × 0.25 mm, 0.25 μm, Agilent Technologies, Santa Clara, CA, USA). Split-less injection mode was employed, with 99.99% He as carrier gas (Sapio, Bergamo, Italy). The hoven was set as described: 65 °C for 2 min, followed by 8 °C/min to 110 °C, then 17 °C/min to 260 °C, holding the temperature for 10 min. The spectra were recording at specific mass of 73, 75, 117, 129, 132, 145, 159, 171, 173, 187, 201, 215, 229, 243, and 257 *m*/*z*, at 70 eV. All the chromatograms were recorded three times. The obtained mass spectra were interpreted by comparing the profiles with the one present in the repository of the National Institute of Standards and Technology (NIST), and with injected standard molecules (Sigma, Milano, Italy).

### 4.6. Cell Lines and Their Maintenance

CCD841 CoN cells (cell line ATCC, CRL-1790, Manassas, VA, USA), that were isolated from normal human colon tissue, and HT-29 human colorectal adenocarcinoma cells (cell line ATCC, HTB-38, Manassas, VA, USA), as a tumor model for colorectal cancer were used. The normal cultural conditions are reported in De Giani et al. [5].

### 4.7. Viability Assay

MTT cell viability assay [46] was carried out to evaluate whether cells were sensitive to treatment. Cells were prepared as reported in De Giani et al. [45]. Then, they were exposed to 0, 0.1, 0.5, and 1 mg/mL of total extract ginseng (ET), fermented Ginseng extract from the beneficial probiotic mix (EF) or mMRS medium fermented by the beneficial probiotic mix (mMRS). After 48 h at 37 °C, in each well 100 µL of a medium without phenol red replaced the original one, and 10 μL of 5 mg/mL MTT [3-(4,5-dimethylthiazol-2)-2,5-diphenyltetrazolium bromide] solution (Sigma, St. Louis, MO, USA) were added. At the end of an incubation of 4 h for CCD841 or 2 h for HT-29, 100 µL of a solution with 10% Triton-X-100 (Sigma, Milano, Italy) in acidic (CH_3_)_2_CHOH (0.1 N chloride acid, Sigma, Milano, Italy) solubilized the formazan crystals formed. Then, the absorbance was measured at 570 nm. Vitality was indicated as %. 

### 4.8. Antioxidant Enzyme Assays

First, cell lines were seeded at 10^6^ cells in 100 mm Ø cell culture Petri dishes (Euroclone, Pero, Italy) and one day later were treated with 0.5 mg/mL of ET or EF for 48 h. The cell lysis took place in ice with a specific buffer (50 mM Tris-HCl pH 7.4, 150 mM NaCl, 5 mM EDTA, 10% glycerol, 1% NP40 buffer, 1 μM leupeptin, 2 μg/mL aprotinin, 1 μg/mL pepstatin and 1 mM PMSF, all purchase from Sigma, Milano, Italy). Homogenates passed a blunt 20-gauge needle fitted to a syringe (Nipro, Assago, Italy), then subjected to 15,000 rpm centrifuge for 30 min at 4 °C. The enzymatic activities were measured in the obtained supernatants. The activity of catalase (CAT), glutathione S-transferase (GST), glutathione peroxidase (Gpox), glutathione reductase (GR), and superoxide dismutase (SOD) was assayed according to Oldani et al. [47]. CAT activation was measured at 240 nm in line with Bergmeyer et al. [48]; GST was assayed in accordance with Habig et al. [49] at 340 nm; Gpox activity was monitored thanks to the coupling to the reaction of NADPH disappearance operated by GR, detected at 340 nm according to Nakamura et al. [50]; GR activity was coupled to the NADPH disappearing at 340 nm [51]. SOD was measured using Vance et al. [52] indirect assay, at 550 nm. Enzymatic activities were correlated to total protein concentration measured with Bradford method [39].

### 4.9. Glutathione Assay

After the cell seeding at the same density described above and the 48 h treatment with 0.5 mg/mL of ET or EF, cells were harvested and subjected to centrifuge at 1200 rpm (Eppendorf, Milano, Italy) for ten min. The obtained cell pellets were resuspended in 300 μL PBS (Euroclone, Pero, Italy), centrifuged, and weighed to report the results to the amount of obtained cells (expressed as mg). Then, the pellets were treated as reported by Oldani et al. [47]. The results were showed as nmol/mg cells.

### 4.10. Intracellular ROS Detection

The oxidation of 2′,7′-Dichlorofluorescin diacetate (H_2_DCF-DA) (Sigma, St. Louis, MO, USA) [53], Dihydroethidium (DHE) (Sigma, St. Louis, MO, USA) [54], or MitoPY1 (Tocris Bioscience, Bristol, UK) [55] were used to measure the level of ROS formed in the cells. H_2_DCF-DA indicates the presence of both ROS and nitrogen monoxide (•NO); DHE is specific to O_2_•^−^, and MitoPY is used to analyze the mitochondrial hydrogen peroxide production in intact adherent cells. In all experiments, 10^4^ cells per well were plated into a 96 multiwell (Euroclone, Pero, Italy) and exposed to 0.5 mg/mL of ET or EF for 48 h. Samples were incubated with H_2_DCF-DA, MitoPY1 (at a final concentration in PBS of 5 μΜ final), or DHE (at a concentration of 10 μΜ in complete medium) for 20 min in the dark, at 37 °C. After staining, fluorescence intensity of each well was recorded by a microplate reader (Victor, Elx 800, Milano, Italy). The intensity of the fluorescence of DCF and MitoPY1 was detected at λ excitation 485 nm and λ emission 530 nm; HE fluorescence was detected at λ excitation 490 nm and λ emission 585 nm. The fluorescence intensity was normalized to protein concentration (determined employing the method of Bradford, [39]). 

### 4.11. Statistical Analysis

Regarding the measurements of bacterial growth, the results were presented as mean values ± standard error (SE) of each experiment repeated three times. The significance of the results was verified employing t-Student’s test, indicating as * *p*-value < 0.05 or ** *p*-value < 0.01, *** < 0.001, and **** < 0.0001.

The calculation of the Prebiotic Index (PI) was obtained according to Palframan et al. [30].

Concerning the work with the cell lines, each experiment for each concentration was carried out in triplicate; the results are shown as the mean of the independent experiments ± standard error (SE). The statistical significance of the results was verified by Dunnett’s test, indicating * *p*-value < 0.05 or ** *p*-value < 0.01.

## Figures and Tables

**Figure 1 ijms-24-00373-f001:**
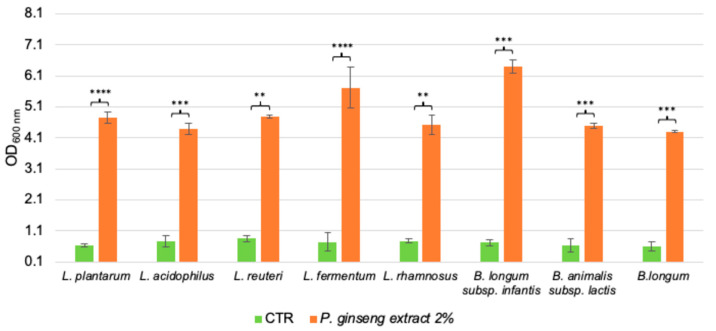
Prebiotic potential of *P. ginseng* berries extract (2% *w*/*v*) on probiotics belonging to *Lactobacillus* and *Bifidobacterium* genera. Growths are represented as OD_600nm_ ± SE. The statistically significant variations are assessed by t-Student’s tests vs. control medium mMRS (CTR) condition of each strain and are depicted as **** *p*-value < 0.0001, *** < 0.001, ** < 0.01.

**Figure 2 ijms-24-00373-f002:**
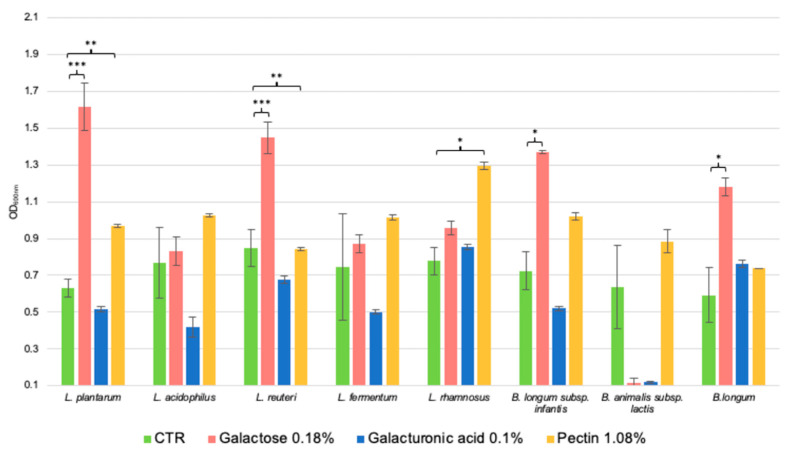
Impact of the single components of *P. ginseng* berries extract on the growth of the selected probiotic strains. Growth values are represented as OD_600nm_ ± SE. The statistically significant results are assessed by t-Student’s tests vs. CTR of each strain and are represented as *** *p*-value < 0.001, ** < 0.01, * < 0.05.

**Figure 3 ijms-24-00373-f003:**
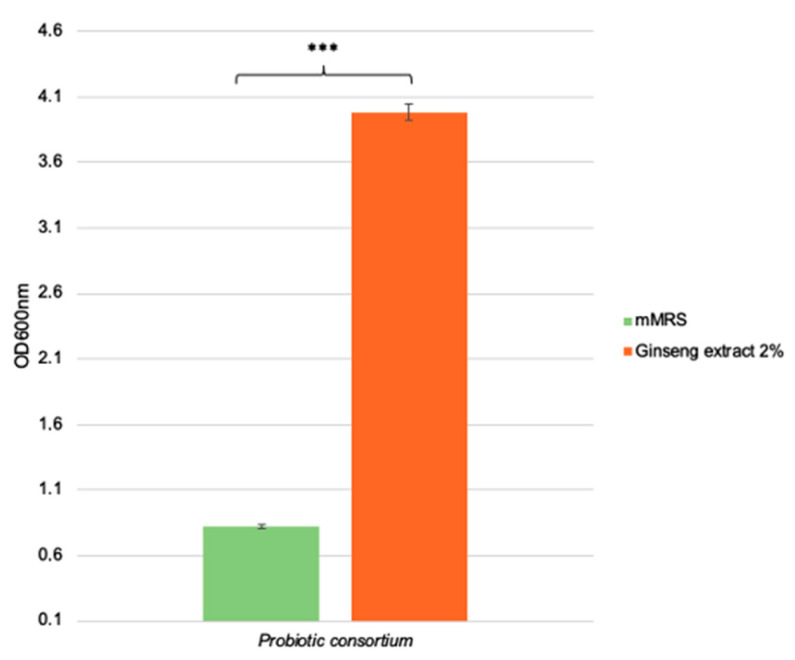
Prebiotic effect of *P. ginseng* berries extract (2% *w*/*v*) on the probiotic consortium. The growth values are represented as OD_600nm_ ± SE. The statistical difference is calculated by t-Student’s tests vs. CTR condition, represented as *** *p*-value < 0.001.

**Figure 4 ijms-24-00373-f004:**
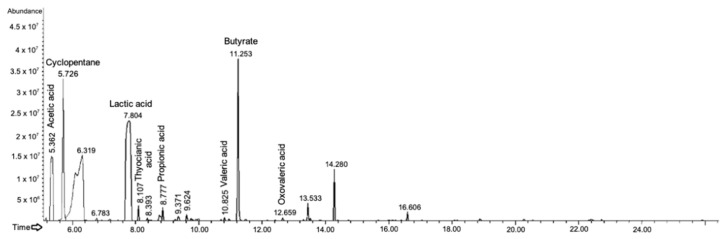
Gas-chromatographic profile of probiotic consortium secondary metabolites after growing on *P. ginseng* berries extract. The names of the principal detected molecules are indicated.

**Figure 5 ijms-24-00373-f005:**
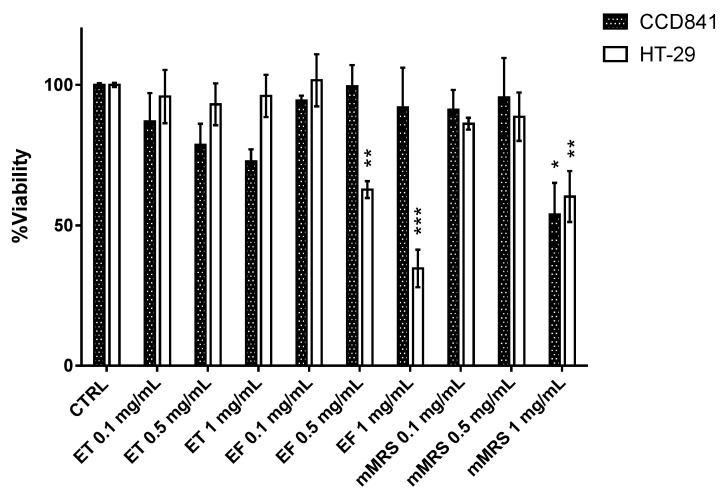
Viability of CCD841 and HT-29 exposed to total Ginseng berries extract (ET), fermented Ginseng berries extract from the probiotic consortium (EF), or mMRS medium fermented by the probiotic consortium (mMRS) at 0.1, 0.5, and 1 mg/mL. Values are depicted as mean viability % ± SE. The statistical significance of the results is calculated employing Dunnett’s multiple comparisons tests: *** *p*-value < 0.001, ** < 0.01, * < 0.05.

**Figure 6 ijms-24-00373-f006:**
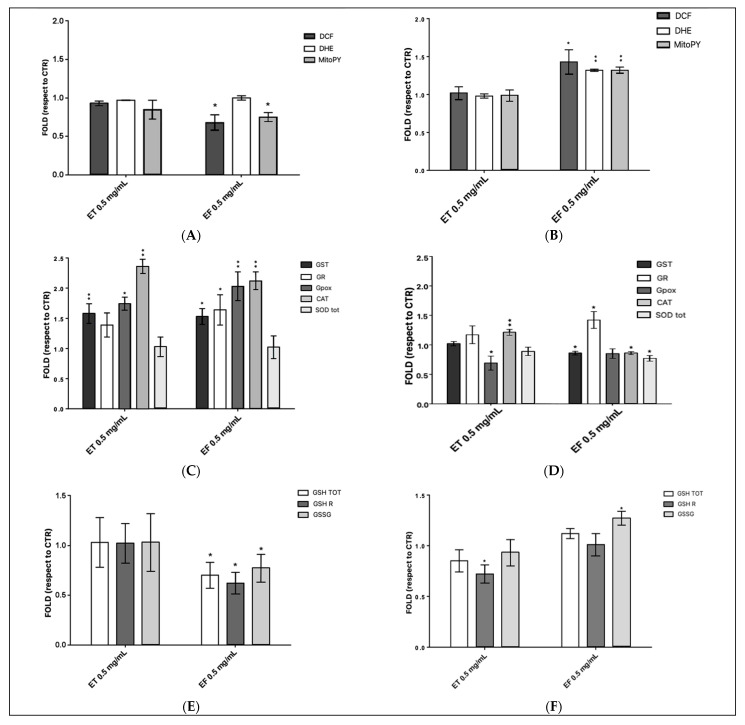
Oxidative stress response of CCD841 and HT-29 after the exposition to total Ginseng berries extract (ET), and fermented Ginseng berries extract from the probiotic consortium (EF) used at 0.5 mg/mL. Panel (**A**) depicts the mean fold value ± SE of generic ROS (measured by H_2_DCF-DA), O_2_•^−^ (measured by DHE), and H_2_O_2_ (measured by MitoPY) in CCD841, (**B**) in HT-29 with respect to the CTR condition. (**C**) represents the fold differences ± SE of the activity of glutathione S-transferase (GST), glutathione reductase (GR), glutathione peroxidase (GPox), catalase (CAT), and total superoxide dismutase (SOD tot) in CCD841, (**D**) in HT-29 with respect to the CTR condition. Finally, (**E**) shows the fold differences ± SE in the levels of total (GSH tot), reduced (GSH), and oxidized (GSSG) glutathione in CCD841, (**F**) in HT-29 with respect to the CTR condition. Statistically significant results are highlighted with ** *p*-value < 0.01, and * < 0.05, calculated employing Dunnett’s multiple comparisons tests.

**Table 1 ijms-24-00373-t001:** *Panax ginseng* berries extract composition. Values are represented as % g/100g of powder.

Component	*P. ginseng* Extract (%)
Pectin-based polysaccharides Galactose Galacturonic acid	54.30% 9.01% 4.84%
Ginsenosides Ginsenosides Re	10.00% 2.00%
Proteins	0.98%
Polyphenols	0.72%
Unidentified molecules	34%

**Table 2 ijms-24-00373-t002:** Bacteria utilized in the study.

Strain	Source	Abbreviation
*Lactobacillus acidophilus* PBS066 (formerly DSM 24936)	Human	LA
*Limosilactobacillus fermentum* PBS072 (formely DSM 25176)	Human	LF
*Lactiplantibacillus plantarum* PBS067 (formely DSM 24937)	Human	LP
*Limosilactobacillus reuteri* PBS073 (formely DSM 25175)	Human	LR
*Lacticaseibacillus rhamnosus* PBS079 (formerly DSM 25568)	Human	LRh
*Bifidobacterium animalis* subsp. *lactis* PBS075 (formerly DSM 25566)	Human	BL
*Bifidobacterium longum* subsp. *longum* PBS108 (formerly DSM 25174)	Human	BLg
*Bifidobacterium longum* subsp. *infantis* LMG P-29639	Human	BI
*Bacteroides cellulosilyticus* CL02T12C19, HM-726	Human	BC
*Bacteroides finegoldii* CL09T03C10	Human	BF
*Clostridium symbiosum* WAL-14673, HM-319	Human	CS
*Clostridium orbiscindens* 1_3_50AFAA, HM-303 (formerly *Flavonifractor plautii*)	Human	CO
*Ruminococcus gnavus* CC55_001C	Human	RG
*Escherichia coli* ATCC 25922	Human	EC

## Data Availability

All data generated during this study are included in this article.

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
