# Peer review of "Synergistic Antioxidant Effect of Prebiotic Ginseng Berries Extract and Probiotic Strains on Healthy and Tumoral Colorectal Cell Lines"

_ijms, 2022, doi:10.3390/ijms24010373_

Round 1

Reviewer 1 Report

ijms-2074768-peer-review-v1

Synergistic antioxidant effect of prebiotic Ginseng berries extract and probiotic strains on healthy and tumoral colorectal 3 cell lines

The work describes the study of the potential antioxidant of an fermented extract obtained from ginseng berries. The chemical composition and the possible antioxidant effect on two types of intestinal cells (CCD841 and HT-29) are analyzed. The work is described in a precise way and has the novelty of analyzing the possible synergistic effect with probiotic microorganisms.

This manuscript is suitable for publication, but minor revisions are described for its better understanding.

Accepted with minor revisión.

Authors should consider the following changes.

In the abstract: The genus of the plant must be described for the first time. In line 12, P. gingeng should be replaced by Panax gingeng.

In Materials and Methods section:

-          In 2.3 Bacterial strains and culture conditions section (page 5), the origin or commercial house of the substances added to the culture of microorganisms (pectin, galactose and galacturonic acids) must be indicated.

-          In 2.7 Viability assay subsection (page 6), the reference of the method followed must be included.

-          In 2.10. Detection of intracelular reactive Oxygen species subsection (page 7), the reference of the methodology followed must also be included.

In Results section:  

In 3.2. Evaluation of the prebiotic potential of the Panax ginseng berries extract subsection (lines 381-384). It has been noted that reference to F. plautii is made, however it does not appear in Table S1. While in table S1 C. orbiscindens appears. The authors must make it clear which microorganisms have been studied.

In Figures 1 and 2 authors should indicate that the concentration of extract was tested at a concentration of 2% (w/v). In addition, in figure 1 it is necessary to indicate the name of the Y axis.

In Figure 6 the same scales must be used to compare the graphs of CCD841 line and HT29 line.

And the figures must have the same size to be able to make a correct comparison.

Author Response

REVIEWER 1

The work describes the study of the potential antioxidant of an fermented extract obtained from ginseng berries. The chemical composition and the possible antioxidant effect on two types of intestinal cells (CCD841 and HT-29) are analyzed. The work is described in a precise way and has the novelty of analyzing the possible synergistic effect with probiotic microorganisms.

This manuscript is suitable for publication, but minor revisions are described for its better understanding.

Accepted with minor revision.

Authors should consider the following changes.

In the abstract: The genus of the plant must be described for the first time. In line 12, P. gingeng should be replaced by Panax gingeng.

Reply: P. ginseng was replaced by Panax ginseng as suggested.

In Materials and Methods section:

-          In 2.3 Bacterial strains and culture conditions section (page 5), the origin or commercial house of the substances added to the culture of microorganisms (pectin, galactose and galacturonic acids) must be indicated.

      Reply: They were indicated in the text as suggested by the reviewer.

-          In 2.7 Viability assay subsection (page 6), the reference of the method followed must be included.

      Reply: The indication of the method used was included in the text and in the references (Mosmann T. Rapid colorimetric assay for cellular growth and survival: Application to proliferation and cytotoxicity assays. J. Immunol. Methods, 1983, 65: 55-63. doi: 10.1016/0022-1759(83)90303-4).

-          In 2.10. Detection of intracelular reactive Oxygen species subsection (page 7), the reference of the methodology followed must also be included.

      Reply: The indication of the methods used were included in the text and in the references (Kim, H.; Xue, X. Detection of total reactive oxygen species in adherent cells by 2',7'-Dichlorodihydrofluorescein diacetate staining. J. Vis. Exp., 2020, 160: e60682. doi:10.3791/60682.; Georgiu, C.D.; Papapostolou, I.; Patsoukis, N.; Tsegenidis, T.; Sideris, T. An ultrasensitive fluorescent assay for the in vivo quantification of superoxide radical in organisms. Anal. Biochem., 2005, 347: 144-151. doi: 10.1016/j.ab.2005.09.013.; Dickinson, B.; Lin, V.; Chang, C. Preparation and use of MitoPY1 for imaging hydrogen peroxide in mitochondria of live cells. Nat. Protoc., 2013, 8: 1249-1259. doi: 10.1038/nprot.2013.064).

In Results section:  

- In 3.2. Evaluation of the prebiotic potential of the Panax ginseng berries extract subsection (lines 381-384). It has been noted that reference to F. plautii is made, however it does not appear in Table S1. While in table S1 C. orbiscindens appears. The authors must make it clear which microorganisms have been studied.

Reply: C. orbiscindens was reclassified as F. plautii (Carlier JP, Bedora-Faure M, K’Ouas G, Alauzet C, Mory F. Proposal to unify Clostridium orbiscindens Winter et al. 1991 and Eubacterium plautii (Seguin 1928) Hofstad and Aasjord 1982, with description of Flavonifractor plautii gen. nov., comb. nov., and reassignment of Bacteroides capillosus to Pseudoflavonifractor capillosus gen. nov., comb. nov. Int J Syst Evol Microbiol. 2010, 60: 585-590). We corrected the name in the text and in table S1.

- In Figures 1 and 2 authors should indicate that the concentration of extract was tested at a concentration of 2% (w/v). In addition, in figure 1 it is necessary to indicate the name of the Y axis.

Reply: We improved the figures indicating the concentration of the extract and the name of Y axis (Optical density at 600 nms). Both the figures were replaced.

- In Figure 6 the same scales must be used to compare the graphs of CCD841 line and HT29 line. And the figures must have the same size to be able to make a correct comparison.

Reply: We have corrected the figures, also adjusting the scales. The new version of figure 6 was inserted in the text.

Reviewer 2 Report

The study proposed by the authors presents an interesting area of both medicine and natural therapies, which causes interest for both scientists and practitioners.

I am satisfied with the presented publication, which is interesting and written in a good style. 

The authors, no doubt, did a good job, including the application of modern methods and statistic studies in this research. The undoubted advantage of the manuscript is the specific goal, the use of study subject that is considered as by-product of ginseng root culture -berries and interesting and reliable introduction of the presented research. However, while reading the article, only one remark arised, by answering which the authors will improve the presentation of the results.

1. Introduction

Line 68-70 it would be helpful if the chemical structures of main ginsenosides was presented.

2

Point 2.2.4. Determination of ginssnosides content

Please indicate the name and the country of origin of the HPLC analytical equipment.

Author Response

REVIEWER 2

The study proposed by the authors presents an interesting area of both medicine and natural therapies, which causes interest for both scientists and practitioners.

I am satisfied with the presented publication, which is interesting and written in a good style. 

The authors, no doubt, did a good job, including the application of modern methods and statistic studies in this research. The undoubted advantage of the manuscript is the specific goal, the use of study subject that is considered as by-product of ginseng root culture -berries and interesting and reliable introduction of the presented research. However, while reading the article, only one remark arised, by answering which the authors will improve the presentation of the results.

  1. Introduction: Line 68-70 it would be helpful if the chemical structures of main ginsenosides was presented.

Reply:The main characteristics of the chemical structures were included in the text as suggested. (Kim J.H., Yi Y., Kim M., Cho J. Y. Role of ginsenosides, the main active components of Panax ginseng, in inflammatory responses and diseases. J. Ginseng Res., 2017, 41: 435e443. doi: 10.1016/j.jgr.2016.08.004)

  1. Point 2.2.4. Determination of ginssnosides content: Please indicate the name and the country of origin of the HPLC analytical equipment.

Reply:The name and the country of the HPLC apparatus and the analytical equipment were added in the text.
